# The Roles of Gut Microbiome and Plasma Metabolites in the Associations between ABO Blood Groups and Insulin Homeostasis: The Microbiome and Insulin Longitudinal Evaluation Study (MILES)

**DOI:** 10.3390/metabo12090787

**Published:** 2022-08-25

**Authors:** Ruifang Li-Gao, Kirk Grubbs, Alain G. Bertoni, Kristi L. Hoffman, Joseph F. Petrosino, Gautam Ramesh, Martin Wu, Jerome I. Rotter, Yii-Der Ida Chen, Anne M. Evans, Richard J. Robinson, Laura Sommerville, Dennis Mook-Kanamori, Mark O. Goodarzi, Gregory A. Michelotti, Patricia A. Sheridan

**Affiliations:** 1Department of Clinical Epidemiology, Leiden University Medical Center, 2333 ZA Leiden, The Netherlands; 2Metabolon, Inc., Morrisville, NC 27560, USA; 3Department of Epidemiology and Prevention, Wake Forest School of Medicine, Winston-Salem, NC 27101, USA; 4Alkek Center for Metagenomics and Microbiome Research, Department of Molecular Virology and Microbiology, Baylor College of Medicine, Houston, TX 77030, USA; 5School of Medicine, University of California San Diego, La Jolla, CA 92161, USA; 6Department of Biology, University of Virginia, Charlottesville, VA 22904, USA; 7The Institute for Translational Genomics and Population Sciences, Department of Pediatrics, The Lundquist Institute for Biomedical Innovation at Harbor-UCLA Medical Center, Torrance, CA 90502, USA; 8Department of Public Health and Primary Care, Leiden University Medical Center, 2333 ZA Leiden, The Netherlands; 9Division of Endocrinology, Diabetes and Metabolism, Department of Medicine, Cedars-Sinai Medical Center, Los Angeles, CA 90048, USA

**Keywords:** insulin homeostasis, ABO blood type, metabolites, gut microbiome, mediation analysis

## Abstract

Non-O blood groups are associated with decreased insulin sensitivity and risk of type 2 diabetes. A recent study pinpointed the associations between ABO blood groups and gut microbiome, which may serve as potential mediators for the observed increased disease risks. We aimed to characterize associations between ABO haplotypes and insulin-related traits as well as potential mediating pathways. We assessed insulin homeostasis in African Americans (AAs; *n* = 109) and non-Hispanic whites (*n* = 210) from the Microbiome and Insulin Longitudinal Evaluation Study. The ABO haplotype was determined by six SNPs located in the *ABO* gene. Based on prior knowledge, we included 21 gut bacteria and 13 plasma metabolites for mediation analysis. In the white study cohort (60 ± 9 years, 42% male), compared to the O1 haplotype, A1 was associated with a higher Matsuda insulin sensitivity index, while a lower relative abundance of *Bacteroides massiliensis* and lactate levels. Lactate was a likely mediator of this association but not *Bacteroides massiliensis*. In the AAs group (57 ± 8 years, 33% male), we found no association between any haplotype and insulin-related traits. In conclusion, the A1 haplotype may promote healthy insulin sensitivity in non-Hispanic whites and lactate likely play a role in this process but not selected gut bacteria.

## 1. Introduction

The ABO blood groups are encoded by the *ABO* locus located on chromosome 9 (9q34.1-q34.2) and can be phenotypically determined by serological tests or genetically defined by single nucleotide polymorphisms (SNPs) at the *ABO* locus [1]. Group O individuals express the H antigen, the precursor to A and B antigens. Group A individuals have α1,3-N-acetylglucosamine (GalNAc) attached to the galactose residue of the H antigen, whereas Group B individuals have α1,3-Gal [2]. The ABO blood groups have been associated with different disease risks, including venous thrombosis, myocardial infarction and type 2 diabetes (T2D) [2,3,4,5].

Type 2 diabetes develops through many complex interactions between genetic, metabolomic, metabolic and lifestyle traits. One example of such an interaction is the potential interplay between ABO blood groups, the gut microbiome, and insulin homeostasis. Individuals with non-O blood types reportedly have a higher risk of developing T2D [6]. While this association is poorly understood, emerging evidence suggests a role for the gut microbiome in mediating T2D risk in individuals with non-O blood type. Recently, our group showed that decreased insulin secretion in response to a liquid meal positively correlated with an intronic variant, rs505922, located in the *ABO* gene. Our confirmatory experiments showed that knocking down the *ABO* gene in a murine pancreatic β-cell line also decreases insulin secretion [7]. In a separate study, genome-wide association profiling of five German cohorts identified two genetic variants in the *ABO* gene, rs3758348 and rs8176632, that correlated with high relative abundance of the gut microbiota species *Faecalibacterium* UTO99_16 and *Bacteriodes* UTO97_27. That study also showed a positive correlation between non-O blood groups and relative abundance of UTO97_27 [8]. Short-chain fatty acid (SCFA)-producing bacteria, including *Faecalibacterium prausnitzii* and *Bacteroides* spp., have been associated with both ABO blood group [8] and increased risk of T2D [9]. In line with this, a recent study also showed that *ABO* genotype alters the gut microbiota by regulating GalNAc levels in pigs [10]. In addition, a low abundance of *Akkermansia muciniphila* has been associated with prediabetes and newly diagnosed T2D [11]. Similarly, butyrate-producing bacteria (e.g., *Roseburia intestinalis* and *Faecalibacterium prausnitzii*), known for their anti-inflammatory properties [12], were less abundant in subjects with T2D [13,14]. Altogether, these data suggest that the association between non-O blood types and increased risk of T2D may be mediated by the gut microbiome.

An increasing body of evidence suggests that the gut microbiome influences host physiology through the production of small molecules, i.e., metabolites [15]. Circulating metabolites have been shown to reflect the diversity of the gut microbiome and may inform T2D risk [16,17]. Extensive investigations have been performed to identify metabolites that are associated with T2D, and branched-chain amino acids, amongst others, robustly showed their associations to increase the risk of T2D [18,19,20,21,22].

The goals of this study were to further characterize the association between ABO haplotypes and insulin homeostasis and to identify potential biological pathways that mediate this process through the gut microbiome and circulating metabolites. For this reason, we profiled plasma samples for metabolite mediators in addition to profiling fecal samples for gut microbiota. Samples came from individuals recruited to the Microbiome and Insulin Longitudinal Evaluation Study (MILES). This study profiled a cohort of 353 non-Hispanic white and African American individuals without diabetes over a period of 30 months to investigate the role of the gut microbiome in regulating key insulin homeostasis traits that lead to the development of T2D [23]. We profiled samples from this cohort to identify associations between ABO blood groups and insulin-related traits (insulin secretion, insulin sensitivity, disposition index, insulin clearance).

## 2. Methods and Materials

### 2.1. Study Population

This study was performed in the community-based prospective cohort, the Microbiome and Insulin Longitudinal Evaluation Study (MILES) [23]. The study was approved by institutional review boards at participating centers and all subjects gave written informed consent prior to participation. Detailed information about the study design, data collection, and patient population has been described [23]. Briefly, 353 non-diabetic individuals (129 African Americans and 224 non-Hispanic whites) aged 40 to 80 years were enrolled. Participants were recruited in the Piedmont Triad area of North Carolina, including Forsyth, Davidson, Davie, Guilford, Stokes, and Surry counties. Race and ethnicity were self-identified. Participants attended a baseline clinic visit and two follow-up visits approximately 15 months apart, for a total of three visits. Though COVID has altered the timing of some of the collections, only baseline samples were used for this current study. At the baseline visit, anthropometric measurements (weight, height, waist, and hip circumference), a full medication history, and medical history were documented for each subject. Before the baseline visit each subject completed a comprehensive health questionnaire and received stool collection materials by mail. The current study is based on 109 non-Hispanic white and 210 African American participants who had complete microbiome, genetic, and metabolite data.

### 2.2. ABO Haplotype and Diplotype

Genotyping was performed using Illumina Global Screening Array v3 (Illumina Inc., San Diego, CA, USA). Genotypes were further imputed to the Haplotype Reference Consortium (HRC) release 1.1 using Sanger Imputation Service. Individuals were excluded if not within mean heterozygosity ± 3SD (standard deviation). Samples were included if call rate above 95%. Variants were included if Hardy–Weinberg Equilibrium (HWE) *p*-values above 1 × 10^−6^ and minor allele frequency (MAF) ≥ 1%, SNP call rate ≥ 98%. We extracted relevant SNPs from the imputed genome and determined ABO blood group genotyping single nucleotide polymorphisms (SNPs) that tag the serological ABO blood groups. We selected the following six SNPs located at the *ABO* locus to tag ABO haplotypes (1): rs8176719-C insertion for ABO O1, rs1053878-A allele for A2, rs8176743-T allele for B, rs41302905-T allele for O2, and rs579459-C allele and rs2519093-T allele for A1 haplotype. We used two SNPs to tag the A1 haplotype because the rs579459-C allele was shown to be a less than perfect genetic marker [1] so the rs2519093-T allele was used to confirm all A1 haplotypes (Appendix A).

The resulting haplotype structure allows us to determine the pair of haplotypes (i.e., diplotypes) carried by each person. In the white cohort, the diplotypes were determined in 88% individuals without ambiguity. However, less than 50% of diplotypes were inferred in the African American cohort using the six tag SNPs, which led to a smaller sample size (*n* = 55). Therefore, only white participants were included in diplotype analyses.

### 2.3. Insulin Homeostasis Measurements

At baseline visits, all participants drank a 75 g glucose load following an overnight fast to determine their oral glucose tolerance. We drew venous blood samples to measure plasma glucose, insulin, and C-peptide levels at fasting and 30 min and 120 min after drinking the glucose load. We assessed insulin sensitivity according to four indices as previously described (23). Briefly, the Matsuda insulin sensitivity index (ISI) was used to inform insulin sensitivity according to: 10000glucose0×insulin0×glucosemean×insulinmean. Insulin secretion (AUC-Ins_30_/AUC-Glu_30_) was calculated as the area under the curve (AUC) for insulin levels at baseline to 30 min after drinking the glucose load according to: ${\mathrm{AUC}}_{\mathrm{insulin}30}/{\mathrm{AUC}}_{\mathrm{glucose}30}$. This calculation is an accurate proxy for the first spike (i.e., phase) of insulin secretion in response to oral glucose. Third, insulin clearance (AUC-Cpep/AUC-Ins) was measured as the AUC of C-peptide over the AUC of insulin according to: ${\mathrm{AUC}}_{C_{\mathrm{peptide}}}/{\mathrm{AUC}}_{\mathrm{insulin}}$. This calculation measures hepatic insulin extraction because the liver clears insulin but not C-peptide [23]. Finally, the disposition index (DI_30_) was calculated as the product of insulin secretion and insulin sensitivity according to: ISI×AUC−Ins30/AUC−Glu30. This measurement of insulin secretion accounts for its degree of compensation for insulin resistance [23]. For all calculations, we applied natural logarithm transformation to obtain normal distributions.

### 2.4. Sample Extraction and Metabolite Profiling

Baseline plasma samples collected from study participants were sent to Metabolon, Inc. (Durham, NC, USA) for untargeted metabolomic profiling on the Metabolon™ Discovery HD4 platform using published methods [24,25,26]. Samples were diluted with methanol, shaken and subsequently centrifuged to precipitate protein, liberate small molecules associated with or trapped in proteins, and to maximize the diversity of metabolites recovered. The resulting extracts were divided into two for analysis by two separate reverse phase (RP)/ultra-performance liquid chromatography (UPLC)-mass spectrometry (MS)/MS methods with positive ion mode electrospray ionization (ESI), one for analysis by RP/UPLC-MS/MS with negative mode ESI, one for analysis by hydrophobic interaction chromatography/UPLC-MS/MS with negative ion mode ESI, and 4 plates reserved for potential re-analysis if needed. Chromatography conditions are provided in Appendix A. The plates were dried under warm N2 and stored. Immediately prior to LC-MS analysis the individual plates were reconstituted in solvents specific to the designated analytical method (Appendix A). All analyses were performed using Waters Acquity UPLC systems plumbed to allow alternating injections on 2 columns coupled to Thermo Q-Exactive mass spectrometers. Metabolon identified compounds by using proprietary in-house software to compare experimental data to a library of authentic standards containing accurate mass, retention time, and fragmentation data as previously described [24,25,26]. Analyzed and interspersed with the experimental samples were a series of QC samples including technical replicates and process blanks. Process blanks consisted of aliquots of DI H_2_O taken through the entire analytical process while the technical replicates consisted of aliquots of pooled human plasma. These samples, along with the inclusion of both instrument performance standards and process assessment standards (Appendix A) in each sample, were used to confirm analytical performance. The process blanks were further used to remove artifacts from the final data set (artifacts, in this case, being defined as any compound with a signal intensity <3× the signal in the process blanks).

### 2.5. Metagenomics Measurements

Whole metagenome shotgun (WMS) sequencing was performed at the Baylor College of Medicine Alkek Center for Metagenomics and Microbiome Research (CMMR) (23). Sequencing produced 10 gigabases per sample to allow greater coverage of the pangenome. For data processing, we used Casava (Illumina) to generate fastq files and BBDuk1 to trim and filter the data. We performed taxonomic profiling on the final read set using MetaPhlAn3 [27]. Based on previously reported associations between ABO blood type and gut microbiome as well as the associations between the gut microbiome and T2D, the relative abundance of all species of *Bacteroides*, *Faecalibacterium, Roseburia*, and *Akkermansia* were obtained from the output of MetaPhlAn3. Because the relative abundances for many of the individual bacterial species showed an excess of zeros [28], we refined our dataset to contain species with at least 30% non-zero values. This refinement left 21 bacteria species for analysis: *Bacteroides caccae*, *Bacteroides cellulosilyticus*, *Bacteroides dorei*, *Bacteroides faecis*, *Bacteroides finegoldii*, *Bacteroides fragilis*, *Bacteroides galacturonicus*, *Bacteroides massiliensis*, *Bacteroides ovatus*, *Bacteroides stercoris*, *Bacteroides thetaiotaomicron*, *Bacteroides uniformis*, *Bacteroides vulgatus*, *Bacteroides xylanisolvens*, *Faecalibacterium prausnitzii*, *Roseburia faecis*, *Roseburia hominis*, *Roseburia intestinalis*, *Roseburia inulinivorans*, *Roseburia_sp_CAG_471*, *Akkermansia muciniphila*. We applied centered additive log-ratio (CLR) transformation to relative abundance of the bacterial taxon, with the geometric mean calculated on the taxonomic level of species used as denominator [29]. The distributions of CLR transformed relative abundance of the 21 bacterial species were compared using Kolmogorov–Smirnov (KS) test in two ethnicities, a nonparametric method for comparing two samples.

### 2.6. Statistical Analysis

Haplotype association analyses were performed using the haplo.stats package in R [30]. Using best-guessed genotypes obtained from imputed data, haplo.stats implements linear regression models to estimate the effect of haplotype on: (1) insulin-related traits (*n* = 4), (2) levels of circulating metabolites (*n* = 13), and (3) relative abundance of bacteria species (*n* = 21). The program achieves this by comparing study variables with a reference haplotype under the assumption of additive effects. For our analyses, O1 served as the reference haplotype. White and African American groups were analyzed separately. All analyses within each group were adjusted for age, sex and the first ten principal components. Due to the hypothesis-driven nature of the analyses, we considered a nominal *p* value threshold of 0.05 to be statistically significant. A more stringent statistical significance was set based on the Bonferroni correction of 0.05/n_traits_ as follows: insulin-related traits, 0.05/4 = 0.01; metabolites, 0.05/13 = 0.004; bacterial species, 0.05/21 = 0.0024. In the white study cohort, we tested the associations between ABO diplotypes and insulin-related traits, metabolite levels, and relative abundance of bacterial species using a linear regression model, adjusted for the same covariates as for the haplotype analyses and adopted the same significance levels.

For the significant associations between ABO haplotype/diplotype and insulin-related traits, we performed a mediation analysis using the mediation package in R [31]. We checked the fulfilment of the four assumptions of the Baron-Kenny framework [32], by assessing the associations between: (i) ABO haplotype/blood type and insulin-related traits with linear regression analyses (total effect C); (ii) ABO haplotype/blood type and mediator with linear regression analyses (indirect effect A); (iii) mediator and insulin-related traits with linear regression analyses (indirect effect B); and (iv) if the association between ABO haplotype/blood type and insulin-related traits attenuated after adding the mediator to the model (direct effect C’). Fulfilment of the assumptions was based on the size of the effect estimate in the regression analysis rather than statistical significance. Average direct effects (ADE) are defined as the direct effects of ABO haplotype/blood type on insulin-related traits, while average causal mediation effects (ACME) describe the effects of ABO haplotype/blood type on insulin-related traits that go through a mediator. The sum of ADE and ACME is total effect, and the proportion of mediation is calculated by ACME accounting for a total effect. For mediators that were associated with both an exposure and an outcome and reached a nominal *p* < 0.05, we reported the average causal mediation effects and the proportion of a total effect explained by a mediator.

## 3. Results

The study cohort consisted of 319 individuals from two ethnic groups, with 210 self-reported non-Hispanic white participants and 109 self-reported African American participants (Table 1). The average ages of white and African American participants were 60 and 57 years, respectively. Women were overrepresented in both ethnic groups. The average body mass index (BMI) was higher in the African American group (32.5 kg/m^2^ vs. 27.3 kg/m^2^). Among the six SNPs used to tag ABO haplotypes, all of them were obtained from imputed genotype data, with imputation quality between 0.92 and 0.99. rs8176719 (used to tag ABO O1) and rs579459 (used to tag A1) were not present and were perfectly tagged by rs8176645 and rs600038 separately, with linkage disequilibrium (LD) between 0.98 and 0.99. All SNPs passed the Hardy–Weinberg equilibrium (HWE) tests with *p*-values > 0.05.

In the first set of studies, we assessed the associations between ABO haplotypes and insulin-related traits. In the white group, the six *ABO* SNPs generated four common haplotypes with frequencies ranging between 68% (O1) and 6% (B) (Table 2). Compared to the O1 haplotype reference group, individuals with haplotype A1 had higher ISI values (β [95% confidence interval]: 0.132 [0.049, 0.216]; *p*-value < 0.01), which is equivalent to a 14.1% increase of ISI after back-transformation from natural logarithm transformation of ISI. Individuals with A2 haplotype also demonstrated nominally higher levels of ISI (0.137 [0.023, 0.251]; 0.01 < *p*-value < 0.05) and AUC-Cpep/AUC-Ins (0.066 [0.01, 0.123]; 0.01 < *p*-value < 0.05) compared to the O1 reference group. This is equivalent to a 14.7% increase of ISI and 6.8% increase of AUC-Cpep/AUC-Ins after back-transformation from natural logarithm transformation of the insulin-related traits. In the African American group, the six *ABO* SNPs yielded four common haplotypes (namely A1, A2, B and O1) and two unknown haplotypes. We found no association between any of the haplotypes and any insulin related trait in this group (Table 2).

To identify biological pathways that may mediate the association between haplotypes and insulin homeostasis, we analyzed the relative abundance of 21 gut bacterial species and 13 circulating metabolites in A1 haplotype samples. A total of 1253 named metabolites were identified in plasma, including 207 amino acids, 24 carbohydrates, 41 cofactors and vitamins, 9 energy-related metabolites, 579 lipids, 35 nucleotides, 26 partially characterized molecules, 66 peptides and 266 xenobiotics. To balance the sample size limitation and the depth of omics data profile in the MILES study, we set up the current study as a hypothesis-driven analysis and selected 13 metabolites that have been associated with glucose metabolism, insulin homeostasis and/or T2D [33,34,35], namely isoleucine, leucine, valine, lactate, glucose, 1,5-anhydroglucitol, 2-hydroxybutyrate, N-lactoyl phenylalanine, N-lactoyl tyrosine, N-lactoyl valine. N-lactoyl leucine, N-lactoyl isoleucine and metabolonic lactone sulfate (previously X-12063). The mass accuracy for these metabolites is included in Appendix A. In the white group, individuals with A1 haplotype showed a lower relative abundance of *Bacteroides massiliensis* (−2.02 [−3.3, −0.75]; *p*-value < 0.0024) (Table 3) and a low level of lactate (−0.092 [−0.181, −0.002]; 0.004 < *p*-value < 0.05) (Table 4), showing that both factors may mediate these associations. To further characterize the potential role of *Bacteroides massiliensis* and lactate in regulating insulin sensitivity, we performed a mediation analysis. While we found no association between relative abundance of *Bacteroides massiliensis* and ISI, we found a strong association between ISI and lactate levels (*p*-value < 0.05). Our mediation analysis showed that lactate accounted for 24.2% of the total effect between A1 haplotype and ISI (Figure 1), indicating that lactate is likely to play a role in mediating insulin homeostasis in white individuals with A1 haplotype.

Although we found no associations between ABO haplotypes and insulin homeostasis in the African American population, we identified several significant associations (*p*-value < 0.05) between ABO haplotypes and the relative abundance of bacterial species (Table 3). Specifically, haplotype A1 showed a significant association with the relative abundance of *Akkermansia muciniphila* in African Americans (4.44 [1.7, 7.17], *p*-value < 0.0024). These results may be explained by distinctive distributions of relative abundance between two ethnic groups (Appendix A).

To further characterize associations between ABO group and insulin homeostasis, we analyzed diplotypes. Among 210 white participants in the MILES study, we were able to unambiguously identify diplotypes in 185 individuals. In this group, 100 people had blood type O, 62 blood type A, 18 blood type B, and 5 blood type AB. Compared to the blood type O reference group, individuals with blood type A had higher levels of ISI (0.18 [0.088, 0.28]; *p*-value < 0.01) and AUC-Cpep/AUC-Ins (0.070 [0.021, 0.12]; *p*-value < 0.01) (Figure 2A). We tested all associations between ABO blood groups, the relative abundance of bacterial species, and circulating metabolites (Figure 2B,C). These studies showed that blood type A is associated with low levels of N-lactoyl leucine and metabolonic lactone sulfate (Figure 2C). Interestingly, both N-lactoyl leucine and metabolonic lactone sulfate were also strongly associated with ISI and AUC-Cpep/AUC-Ins (Figure 3A,B). A mediation analysis showed that in individuals with blood type A, N-lactoyl leucine and metabolonic lactone sulfate explained 25.3% and 21.4% of the total effect (0.182) on ISI, respectively (Figure 3A). Similarly, N-lactoyl leucine and metabolonic lactone sulfate explained 22.9% and 18.6%, respectively, of the total effect (0.070) on AUC-Cpep/AUC-Ins (Figure 3B).

## 4. Discussion

This study examined the associations between ABO haplotypes and ABO blood groups with insulin homeostasis in non-Hispanic whites and African Americans. The A1 haplotype was associated with higher ISI in the White group, indicating healthier insulin sensitivity in these individuals. Untargeted metabolic profiling, statistical analyses, and follow-up mediation analysis suggest that plasma lactate levels may mediate this association. Furthermore, in the white population, blood type A showed a strong association with higher ISI and higher insulin clearance compared to the blood type O control group. This suggests that individuals with blood type A may be at a lower risk of developing T2D. Both associations may be mediated by N-lactoyl leucine and metabolonic lactone sulfate levels in plasma.

ABO blood groups have been linked to the risk of developing various diseases, with recent attention focusing on the association between ABO blood groups and type 2 diabetes [36]. Three independent phenome-wide association studies (PhWAS) have recently been performed with the goal of systematically quantifying the associations between ABO blood groups and the risk of developing cardiometabolic diseases, including T2D [3,4,37]. Overall, all three studies suggested a null association between the ABO blood group and T2D. Specifically, the first study found no association between the ABO blood group and T2D risk. It did, however, find that O and B antigens were protective for glucose metabolism while A1 was associated with a higher level of fasting glucose (4). The second study reported only a weak association between the ABO blood group and T2D risk that did not reach significance after multiple testing corrections (non-O vs. O (β [95% confidence interval]): 1.051 [1.017–1.086], *p*-value = 3.02 × 10^−3^) [3]. The third study profiled all ~500,000 participants in the UK Biobank [37]. The first data analysis included only participants who identified as white British. The second data analysis included all participants regardless of their ethnicity. The analysis that included all participants showed an association between blood type O and a lower risk of T2D. However, this association did not remain significant when restricting the analysis to the white British population. As this change in results may be explained by either a decreased sample size or removing a confounding issue from population stratification, these data are inconclusive [37]. However, profiling of the entire study population did reveal a significant association between blood type B and a higher risk of type 1 diabetes (B vs. O (β [95% confidence interval]): 1.78 [1.28–2.39], *p* = 0.001) [37].

Our findings associating A1 haplotype and blood type A with higher ISI may seem inconsistent with the null findings of blood types and T2D risk observed in the UK Biobank. It is worth noting that individuals in the UK Biobank with blood type A/haplotype A1 exhibited inverse associations to cofactors of insulin resistance (e.g., high blood pressure, low high-density cholesterol levels, and high triglycerides), compared to blood type O/haplotype O1 [3,4,37]; positive associations were observed for fasting glucose levels and low-density lipoprotein cholesterol (LDL-C). By contrast, a positive association with high-density lipoprotein cholesterol (HDL-C) and negative associations were observed for triglycerides (TG) and diastolic blood pressure (DBP) [3,4]. Taken together, individuals with blood type A/haplotype A1 may tend to have higher LDL-C levels and higher HDL-C levels with corresponding lower TG levels compared to blood type O/haplotype O1, which made the association to insulin sensitivity elusive based on the evidence from UK Biobank. Yet, even with a beneficial insulin profile found in the current study, individuals with A blood type did not show a lower risk of developing T2D, due to different subtypes of this disease. Mild obesity-related diabetes (MOD) and mild age-related diabetes (MARD) account for more than 40% of total T2D cases and for these subtypes, insulin resistance may not pronouncedly manifest during the progression of the disease compared to the other clusters hallmarked by insulin resistance [38]. Therefore, further studies are needed to examine the associations between ABO blood groups and the risk of T2D specifically in the subtypes that do exhibit insulin resistance.

Through mediation analyses, we found that lactate may mediate the association between haplotype A1 and insulin sensitivity, and that N-lactoyl leucine and metabolonic lactone sulfate may mediate the link between blood type A and insulin sensitivity. In humans, lactate is a product of anaerobic glycolysis, produced minimally and in large part in hypoxic states by myocytes and erythrocytes. Our finding suggested that there may be a common upstream component, perhaps in the *ABO* gene itself, that predisposes an individual to a lower lactate state (i.e., less anaerobic glycolysis) as well as coding for the blood type. N-lactoyl leucine represents the modification of the common amino acid leucine, which is directly related to lactate levels in the sample. When lactate increases the modified amino acid also tends to increase, implying that the levels of these two metabolites are regulated by the same mechanism. A previous study showed that N-lactoyl leucine is formed from lactate and leucine by reversed action of the protease cytosolic nonspecific dipeptidase 2 [39]. Though highly speculative, lactoyl-leucine may serve as a “metabolic sink” for lactate and could function to buffer the deleterious effects of lactate on glucose homeostasis. Alternatively, more recent work suggests that lactate may fuel the generation of key lactoyl-amino acid derivatives that biochemically couple our metabolic state with endocrine signaling. Direct dosing of lactoyl-phenylalanine, for example, was able to improve glucose homeostasis and reduce adiposity in a murine diet-induced obesity model, suggesting a role as a key metabolic regulator of glucose utilization and energy expenditure [40]. Similarly, metabolonic lactone sulfate is (previously named X-12063) also deserves further study to better understand its role in the development of T2D. It has been associated with a broad category of cardiometabolic phenotypes including insulin sensitivity and T2D [33,41,42]. Although short-chain fatty acid (SCFA)-producing bacteria, including *Faecalibacterium prausnitzii* and *Bacteroides* spp., have been associated with both ABO blood group [8] and increased risk of T2D [9], no association between either of these taxa and insulin-related traits were demonstrated in this study. These results may be explained by the limited sample size in both ethnic groups.

In summary, our findings support previous studies showing an association between the ABO blood group and insulin homeostasis and offer molecules that may mediate these associations. The main strengths of this study are: (1) the deep phenotyping of the MILES study, which allowed extensive interrogation of insulin related traits and biological pathways in metabolites and gut microbiome, and (2) both African American and non-Hispanic white populations were studied. One limitation of this study was our inability to identify and fully characterize certain ABO subtypes that were specific to the African American population. This issue may be addressed by performing follow-up studies using novel SNPs. In addition, we only incorporated 21 bacterial species and 13 metabolites based on prior knowledge in the mediation analysis, and there could be other potential mediation pathways that did not cover in the current study.

## 5. Conclusions

In the non-Hispanic white population, both A1 haplotype and blood type A were associated with insulin sensitivity indexed by ISI, with evidence of mediation by lactate. N-lactoyl leucine and metabolonic lactone sulfate levels may mediate the association between blood group A and ISI, and the association between blood type A and insulin clearance. Although several *Bacteroides* species were associated with A2 haplotypes in the African American group, no association was demonstrated between ABO haplotypes or blood types and insulin-related traits in this subgroup.

## Figures and Tables

**Figure 1 metabolites-12-00787-f001:**
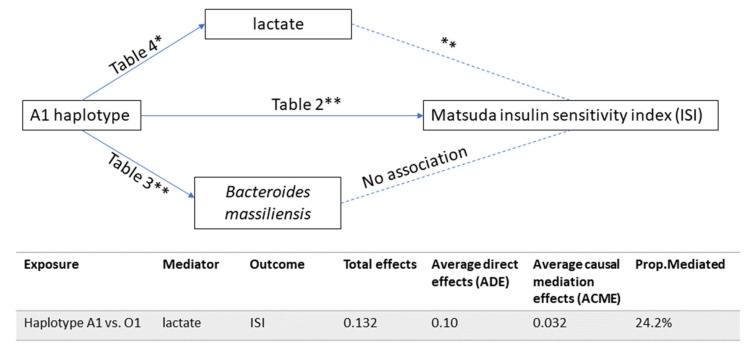
Mediation analysis between A1 haplotype and ISI. * marginally significant with a *p*-value < 0.05. ** Bonferroni corrected *p*-value < 0.004 (0.05/13) for bacterial species, and Bonferroni corrected *p*-value < 0.0024 (0.05/21) for metabolites.

**Figure 2 metabolites-12-00787-f002:**
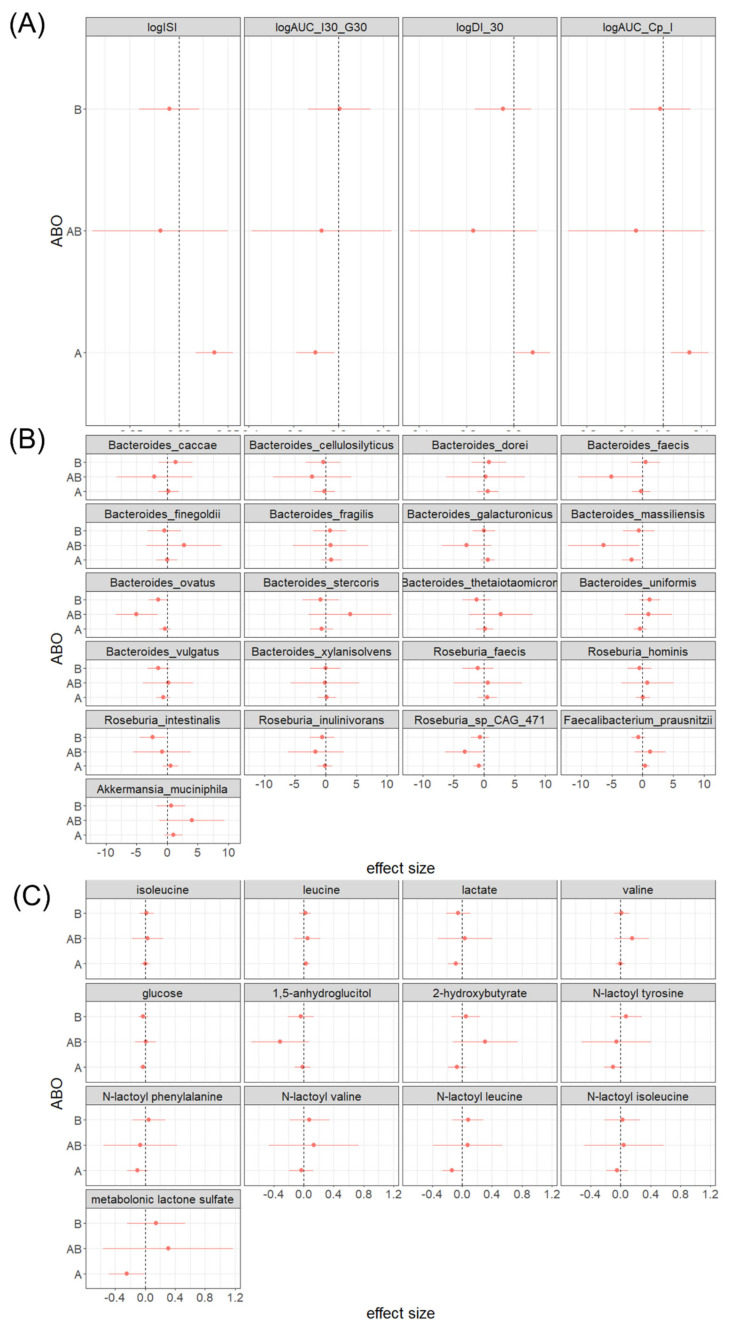
The associations between ABO blood group and (**A**) insulin-related traits, (**B**) relative abundance of selected species as well as (**C**) selected diabetes-associated metabolites in the white population of the MILES study.

**Figure 3 metabolites-12-00787-f003:**
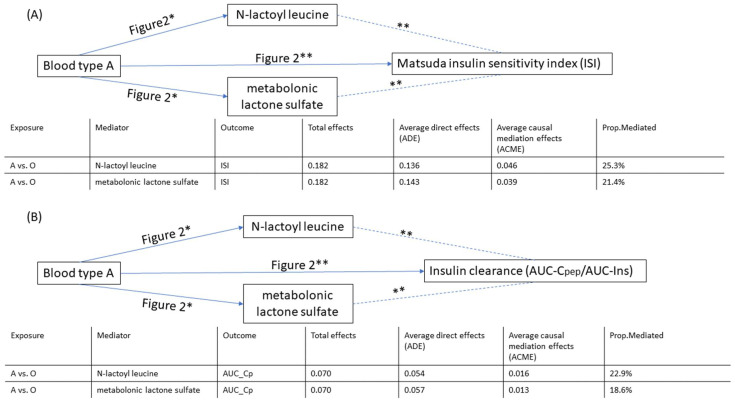
Mediation analysis between blood type A and insulin-related traits. (**A**) illustration of mediation analysis between blood type A and ISI; (**B**) illustration of mediation analysis between blood type A and insulin clearance. * marginally significant with a *p*-value < 0.05. ** Bonferroni corrected *p*-value < 0.0024 (0.05/21) for metabolites.

**Table 1 metabolites-12-00787-t001:** Study population characteristics of MILES study.

	Non-Hispanic Whites	African Americans
N	210	109
Age (years)	60 (9)	57 (8)
Male, n (%)	89 (42%)	36 (33%)
BMI (kg/m^2^)	27.3 (5.8)	32.5 (8.9)
rs41302905 (for O2 haplotype)		
MAF	0.021	0.0091
HWE *p*-value	1	1
Imputation quality	0.92	
rs8176743 (for B haplotype)		
MAF	0.069	0.11
HWE *p*-value	1	1
Imputation quality	0.97	
rs1053878 (for A2 haplotype)		
MAF	0.078	0.21
HWE *p*-value	0.12	1
Imputation quality	0.95	
rs8176645 ^β^ (tagging for rs8176719, for O1 haplotype)		
MAF	0.29	0.32
HWE *p*-value	0.32	0.67
Imputation quality	0.82	
rs2519093 (A1 haplotype)		
MAF	0.15	0.091
HWE *p*-value	0.097	1
Imputation quality	0.99	
rs600038 ^β^ (tagging for rs579459, for A1 haplotype)		
MAF	0.18	0.11
HWE *p*-value	0.052	1
Imputation quality	1.00	

MAF: minor allele frequency; ^β^: tag SNPs with a strong linkage disequilibrium to original SNPs: rs8176645 tagging for rs8176719 (LD = 0.98) and rs600038 (LD = 0.99) tagging for rs579459.

**Table 2 metabolites-12-00787-t002:** The associations between ABO haplotypes and insulin related traits in the MILES study.

Haplotype	rs41302905	rs8176743	rs1053878	rs8176645 ^ψ^	rs2519093	rs600038 ^ψ^	frequency	ISI β (95% CI)	AUC-Ins_30_/AUC-Glu_30_β (95% CI)	AUC-Cpep/AUC-Insβ (95% CI)	DI_30_ β (95% CI)
**Whites**											
**O1**	C	C	G	delG	C	T	0.68	Reference			
**A1**	C	C	G	G	T	C	0.13	**0.132 (0.049, 0.216) ****	−0.069 (−0.14, 0.003)	0.037 (−0.005, 0.078)	0.062 (−0.003, 0.127)
**A2**	C	C	A	G	C	T	0.068	0.137 (0.023, 0.251) *	−0.055 (−0.152, 0.043)	0.066 (0.01, 0.123) *	0.082 (−0.007, 0.171)
**B**	C	T	G	G	C	T	0.060	−0.079 (−0.209, 0.051)	0.043 (−0.068, 0.155)	−0.032 (−0.097, 0.032)	−0.035 (−0.137, 0.066)
**African Americans**											
**O1**	C	C	G	delG	C	T	0.52	Reference			
**unknown**	C	C	G	G	C	T	0.061	0.092 (−0.144, 0.329)	−0.035 (−0.225, 0.156)	0.097 (−0.018, 0.213)	0.07 (−0.113, 0.253)
**A1**	C	C	G	G	T	C	0.053	0.062 (−0.146, 0.269)	0.028 (−0.159, 0.216)	0.002 (−0.114, 0.118)	0.104 (−0.065, 0.272)
**A2/O1**	C	C	A	delG	C	T	0.083	0.037 (−0.121, 0.195)	−0.089 (−0.249, 0.07)	0.036 (−0.054, 0.127)	−0.033 (−0.187, 0.122)
**A2**	C	C	A	G	C	T	0.10	0.039 (−0.1, 0.178)	0.036 (−0.087, 0.159)	0.037 (−0.044, 0.119)	0.07 (−0.053, 0.194)
**B**	C	T	G	G	C	T	0.064	−0.044 (−0.218, 0.13)	0.01 (−0.151, 0.171)	−0.088 (−0.195, 0.019)	−0.028 (−0.19, 0.134)

* marginally significant with a *p*-value < 0.05. ** study-wise significance with a Bonferroni corrected *p*-value < 0.01 (0.05/4), in bold. ^ψ^: tag SNPs with strong linkage disequilibrium to original SNPs: rs8176645 tagging for rs8176719 (LD = 0.98) and rs600038 (LD = 0.99) tagging for rs579459. β: point effect size estimates; 95% CI: 95% confidence interval of beta estimations. Insulin related traits are natural logarithm transformed, and all analyses retained haplotypes with a frequency above 5%.

**Table 3 metabolites-12-00787-t003:** The associations between ABO haplotypes and relative abundance of selected 21 bacterial species in the MILES study, compared to the reference group haplotype O1.

			African Americans					Whites		
		Haplotype O1	Haplotype unknown β (95% CI)	Haplotype A1 β (95% CI)	Haplotype A2/O1 β (95% CI)	Haplotype A2 β (95% CI)	Haplotype B β (95% CI)	Haplotype A1 β (95% CI)	Haplotype A2 β (95% CI)	Haplotype B β (95% CI)
**Bacteroides**	Caccae β (95% CI)	Reference	0.13 (−3.79, 4.04)	−0.62 (−3.96, 2.72)	−0.2 (−3.35, 2.94)	0.64 (−1.93, 3.21)	−0.32 (−3.63, 2.99)	0.07 (−1.34, 1.48)	−0.2 (−2.14, 1.74)	0.53 (−1.71, 2.77)
	Cellulosilyticus β (95% CI)		−3.07 (−6.12, −0.01) *	−1.27 (−4.24, 1.7)	−0.15 (−2.8, 2.5)	0.9 (−1.43, 3.23)	0.03 (−2.98, 3.05)	−0.14 (−1.59, 1.3)	−0.37 (−2.37, 1.63)	−0.12 (−2.42, 2.18)
	dorei β (95% CI)		−1.06 (−4.9, 2.79)	−0.74 (−4.05, 2.58)	1.11 (−1.92, 4.14)	2.77 (0.22, 5.33) *	−1.54 (−4.79, 1.71)	0.49 (−0.97, 1.95)	0.64 (−1.37, 2.65)	0.43 (−1.9, 2.75)
	Faecis β (95% CI)		−2.04 (−5.3, 1.21)	−0.4 (−3.55, 2.75)	1.89 (−0.98, 4.77)	0.49 (−1.79, 2.77)	−0.34 (−3.23, 2.54)	−0.21 (−1.43, 1.01)	−0.51 (−2.18, 1.15)	−0.02 (−1.97, 1.93)
	Finegoldii β (95% CI)		−0.51 (−3.55, 2.54)	1.31 (−1.45, 4.06)	2.14 (−0.38, 4.65)	−2.94 (−5.01, −0.86) *	0.43 (−2.11, 2.98)	−0.33 (−1.7, 1.05)	0.13 (−1.78, 2.03)	−0.97 (−3.16, 1.23)
	Fragilis β (95% CI)		0.72 (−2.8, 4.24)	−1.15 (−4.29, 1.98)	−0.61 (−3.38, 2.17)	1.28 (−1.04, 3.59)	0.46 (−2.57, 3.49)	1.14 (−0.21, 2.48)	1.4 (−0.45, 3.25)	1.1 (−1.05, 3.25)
	Galacturonicus β (95% CI)		−1.81 (−3.71, 0.08)	−0.97 (−2.77, 0.83)	−1.57 (−3.23, 0.09)	−0.69 (−2.08, 0.71)	0.77 (−1.1, 2.63)	0.5 (−0.39, 1.38)	−0.11 (−1.32, 1.1)	−0.23 (−1.64, 1.18)
	Massiliensis β (95% CI)		−1.53 (−4.82, 1.76)	1.29 (−1.57, 4.14)	3.37 (0.77, 5.97) *	−2.07 (−4.33, 0.19)	−1.15 (−3.99, 1.69)	**−2.02** **(−3.3, −0.75) ****	−0.57 (−2.34, 1.19)	−1.29 (−3.34, 0.75)
	Ovatus β (95% CI)		−1.67 (−4.59, 1.25)	−1.73 (−4.33, 0.88)	−0.94 (−3.27, 1.39)	0.85 (−1.09, 2.8)	0.33 (−2.24, 2.9)	−0.4 (−1.17, 0.37)	0.39 (−0.67, 1.44)	−1.39 (−2.62, −0.15) *
	Stercoris β (95% CI)		2.88 (−1.25, 7.01)	2.47 (−0.97, 5.91)	1.02 (−2.26, 4.29)	−2.49 (−5.2, 0.22)	3.92 (0.26, 7.58) *	−0.57 (−2.1, 0.96)	−0.52 (−2.63, 1.59)	−0.15 (−2.58, 2.28)
	Thetaiotaomicron β (95% CI)		1.04 (−2.1, 4.19)	1.8 (−1.24, 4.84)	−2.31 (−5.37, 0.75)	1.22 (−0.89, 3.32)	−0.81 (−3.57, 1.95)	−0.15 (−1.35, 1.05)	1.35 (−0.3, 2.99)	−0.37 (−2.26, 1.52)
	Uniformis β (95% CI)		0.71 (−2.42, 3.85)	1.97 (−0.84, 4.78)	1.56 (−0.96, 4.08)	0.41 (−1.72, 2.54)	−0.45 (−3.21, 2.32)	−0.67 (−1.57, 0.24)	0.52 (−0.71, 1.76)	1.12 (−0.31, 2.55)
	Vulgatus β (95% CI)		−2.53 (−5.76, 0.71)	−0.86 (−3.6, 1.88)	−2.66 (−4.97, −0.35) *	−1.14 (−3.21, 0.94)	−1.06 (−3.47, 1.35)	0.08 (−0.93, 1.08)	−0.28 (−1.66, 1.11)	−0.49 (−2.08, 1.09)
	Xylanisolvens β (95% CI)		−0.81 (−3.76, 2.14)	0.44 (−2.33, 3.22)	2.94 (0.46, 5.42) *	−1.68 (−3.72, 0.36)	1.36 (−1.42, 4.14)	0.65 (−0.59, 1.9)	0.9 (−0.81, 2.62)	0.6 (−1.39, 2.58)
**Roseburia**	Faecis β (95% CI)		−0.94 (−4.44, 2.56)	0.94 (−2, 3.88)	−0.47 (−3.3, 2.36)	0.44 (−1.87, 2.75)	−3.7 (−6.81, −0.59) *	−0.19 (−1.46, 1.08)	0.47 (−1.28, 2.23)	−0.14 (−2.16, 1.87)
	Hominis β (95% CI)		0.59 (−2.08, 3.25)	−0.35 (−2.71, 2.01)	−3.1 (−5.24, −0.96) *	0.45 (−1.26, 2.16)	−0.09 (−2.33, 2.15)	0.47 (−0.46, 1.4)	−1.48 (−2.75, −0.2) *	−0.18 (−1.66, 1.29)
	Intestinalis β (95% CI)		−3.09 (−6.17, −0.01) *	−0.18 (−2.97, 2.61)	−0.04 (−2.36, 2.28)	−1.27 (−3.21, 0.67)	−0.92 (−3.38, 1.53)	0.17 (−0.89, 1.23)	0.53 (−0.92, 1.97)	−1.76 (−3.45, −0.08) *
	Inulinivorans β (95% CI)		−2.03 (−4.97, 0.92)	−0.18 (−3.13, 2.77)	−1.16 (−3.26, 0.94)	−0.34 (−2.31, 1.64)	−0.75 (−3.04, 1.55)	0.08 (−0.94, 1.1)	−1.71 (−3.11, −0.32) *	−0.68 (−2.3, 0.94)
	sp_CAG_471 β (95% CI)		−1.8 (−3.47, −0.14) *	1.49 (−0.2, 3.18)	−1.68 (−3.18, −0.17) *	−0.25 (−1.52, 1.01)	−1.46 (−3.08, 0.15)	−0.46 (−1.2, 0.29)	−1.01 (−2.03, 0.01)	−0.26 (−1.45, 0.92)
**Faecalibacterium**	Prausnitzii β (95% CI)		0.55 (−1.05, 2.16)	0.3 (−1.14, 1.74)	−0.85 (−2.27, 0.56)	0.31 (−0.89, 1.51)	−0.36 (−1.81, 1.09)	0.33 (−0.23, 0.89)	0.55 (−0.21, 1.32)	−0.26 (−1.16, 0.63)
**Akkermansia**	Muciniphila β (95% CI)		−2.09 (−5.32, 1.14)	**4.44** **(1.7, 7.17) ****	2.31 (−0.34, 4.97)	1.98 (−0.06, 4.01)	3.55 (0.86, 6.24) *	0.69 (−0.56, 1.94)	0.92 (−0.78, 2.63)	0.55 (−1.44, 2.54)

* marginally significant with a *p*-value < 0.05. ** Bonferroni corrected *p*-value < 0.0024 (0.05/21), in bold. β: point effect size estimates; 95% CI: 95% confidence interval. Species relative abundance is natural logarithm transformed, and all analyses retained haplotypes with a frequency above 5%.

**Table 4 metabolites-12-00787-t004:** The associations between ABO haplotypes and selected metabolites in the MILES study.

		Whites			African Americans				
	Haplotype O1	Haplotype A1 β (95% CI)	Haplotype A2 β (95% CI)	Haplotype B β (95% CI)	Haplotype unknown β (95% CI)	Haplotype A1 β (95% CI)	Haplotype A2/O1 β (95% CI)	Haplotype A2 β (95% CI)	Haplotype B β (95% CI)
**isoleucine**	reference	−0.006 (−0.063, 0.05)	0.005 (−0.072, 0.082)	0.047 (−0.041, 0.136)	−0.118 (−0.251, 0.015)	0.005 (−0.12, 0.13)	−0.124 (−0.238, −0.011) *	−0.018 (−0.115, 0.08)	−0.081 (−0.204, 0.043)
**leucine**		0.015 (−0.035, 0.065)	0.018 (−0.05, 0.087)	0.042 (−0.036, 0.12)	−0.075 (−0.198, 0.049)	0.035 (−0.076, 0.145)	−0.102 (−0.207, 0.003)	0.002 (−0.083, 0.087)	−0.069 (−0.184, 0.046)
**lactate**		−0.092 (−0.181, −0.002) *	−0.033 (−0.155, 0.089)	−0.043 (−0.183, 0.096)	−0.002 (−0.253, 0.248)	0.02 (−0.232, 0.272)	−0.06 (−0.271, 0.15)	−0.161 (−0.34, 0.019)	0.03 (−0.197, 0.257)
**valine**		0.007 (−0.051, 0.066)	0.012 (−0.068, 0.093)	0.048 (−0.044, 0.139)	−0.1 (−0.229, 0.029)	−0.007 (−0.134, 0.12)	−0.105 (−0.216, 0.005)	0.032 (−0.06, 0.123)	−0.092 (−0.211, 0.027)
**glucose**		−0.024 (−0.056, 0.008)	−0.035 (−0.079, 0.009)	−0.01 (−0.06, 0.04)	0.076 (−0.011, 0.162)	−0.069 (−0.162, 0.024)	−0.03 (−0.107, 0.047)	−0.026 (−0.092, 0.041)	0.059 (−0.023, 0.141)
**1,5-anhydroglucitol**		−0.027 (−0.124, 0.069)	0.002 (−0.129, 0.133)	−0.004 ( −0.155, 0.147)	0.068 (−0.177, 0.313)	0.047 (−0.164, 0.259)	−0.07 (−0.266, 0.125)	−0.181 (−0.346, −0.016) *	0.083 (−0.127, 0.294)
**2-hydroxybutyrate**		−0.03 (−0.136, 0.075)	−0.085 (−0.229, 0.059)	0.035 (−0.13, 0.199)	−0.168 (−0.489, 0.153)	−0.206 (−0.496, 0.084)	−0.079 (−0.345, 0.188)	−0.166 (−0.382, 0.05)	−0.157 (−0.439, 0.125)
**N-lactoyl phenylalanine**		−0.05 (−0.166, 0.065)	−0.094 (−0.251, 0.062)	0.049 (−0.131, 0.229)	0.001 (−0.288, 0.29)	−0.056 (−0.323, 0.211)	0.013 (−0.223, 0.249)	−0.083 (−0.282, 0.116)	−0.176 (−0.437, 0.085)
**N-lactoyl tyrosine**		−0.097 (−0.223, 0.028)	−0.016 (−0.189, 0.157)	0.038 (−0.157, 0.232)	−0.049 (−0.353, 0.256)	−0.093 (−0.386, 0.2)	−0.002 (−0.261, 0.258)	−0.012 (−0.247, 0.223)	−0.004 (−0.293, 0.284)
**N-lactoyl valine**		−0.045 (−0.192, 0.101)	0.044 (−0.158, 0.245)	0.115 (−0.119, 0.349)	0.003 (−0.356, 0.363)	−0.077 (−0.401, 0.247)	−0.004 (−0.293, 0.285)	−0.001 (−0.247, 0.244)	−0.207 (−0.524, 0.11)
**N-lactoyl leucine**		−0.105 (−0.224, 0.013)	−0.109 (−0.27, 0.053)	0.029 (−0.156, 0.215)	−0.09 (−0.413, 0.232)	−0.075 (−0.365, 0.215)	−0.088 (−0.351, 0.174)	−0.083 (−0.305, 0.138)	−0.235 (−0.521, 0.051)
**N-lactoyl isoleucine**		−0.048 (−0.184, 0.088)	−0.008 (−0.193, 0.177)	0.038 (−0.173, 0.249)	−0.178 (−0.543, 0.187)	−0.152 (−0.474, 0.17)	0.009 (−0.292, 0.309)	−0.028 (−0.273, 0.216)	−0.191 (−0.505, 0.123)
**metabolonic lactone sulfate**		−0.145 (−0.352, 0.063)	−0.199 (−0.48, 0.081)	0.238 (−0.084, 0.56)	−0.146 (−0.606, 0.314)	−0.123 (−0.566, 0.32)	−0.165 (−0.544, 0.214)	−0.264 (−0.576, 0.048)	−0.218 (−0.631, 0.195)

* marginally significant with a *p*-value < 0.05. The values presented in the table are β (95% CI), with β: point effect size estimates and 95% CI: 95% confidence interval. Metabolite levels are natural logarithm transformed, and all analyses retained haplotypes with a frequency above 5%.

## Data Availability

The data are not publicly available because participants did not give consent for the data to be publicly posted. Interested researchers should contact the corresponding author and submit their credentials to the Cedars-Sinai Institutional Review Board for determination of whether if they are eligible to have access to study data. Upon approval, a limited dataset necessary for replication would be provided to the investigator.

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
