# Peer review of "The Roles of Gut Microbiome and Plasma Metabolites in the Associations between ABO Blood Groups and Insulin Homeostasis: The Microbiome and Insulin Longitudinal Evaluation Study (MILES)"

_metabolites, 2022, doi:10.3390/metabo12090787_

Round 1

Reviewer 1 Report (Previous Reviewer 1)

The authors have addressed all my concerns well.

Author Response

Thank you for the reviewer for the comments. 

Reviewer 2 Report (Previous Reviewer 3)

-

Author Response

Thank you for the reviewer for the comment. 

Reviewer 3 Report (Previous Reviewer 2)

The authors answered the question's point by point, and the quality of the paper has been substantially improved.

The authors mentioned in the paper that 1253 metabolites were identified in plasma. This information should be included in the results section instead of the materials and methods section. In addition, the authors put a lot of effort into performing an intensive metabolomics study, and the information of only 13 metabolites is reported. Therefore, it would be great to write a few lines in the results section about what kind of metabolites were identified in this study, their chemical class information, and the number of identified metabolites in each class group (e.g., how many amino acids, carbohydrates, lipids, etc. were found). After this information, the author can justify why they decided to discuss 13 metabolites in this paper.

In addition, in Table S2. HD4 chromatographic method summary and materials and methods section information about the chromatographic columns that were used in this study is missing. Please add this information to the main text's materials and methods section.

Author Response

This manuscript is a resubmission of an earlier submission. The following is a list of the peer review reports and author responses from that submission.

Round 1

Reviewer 1 Report

Peer review of Ruifang Li-Gao et al, the gut microbiome and plasma metabolites mediate differences in insulin homeostasis associated with ABO blood groups

This manuscript analyzed the association among ABO haplotypes, insulin-related traits, and the gut microbiome/plasma metabolites in two populations from the Microbiome and Insulin Longitudinal Evaluation Study (MILES). The results showed that haplotype A1 was associated with a higher Matsuda Insulin Sensitivity Index (ISI)  (p  <0.01), lower relative abundance of Bacteroides massiliensis as well as a lower concentration of lactate. Mediation analysis showed that lactate is a likely mediator of this association, while Bacteroides massiliensis was not associated with ISI, and therefore not likely to be a mediator. Although different populations (from MILES) and multi-omics approaches (microbiome, genetic, and metabolite data), the author should provide more convincing, novel, or differential results (especially in the microbiome) to show the clinical implications/critical findings in this study. Thus, I would not suggest accepting this manuscript.

 My major concerns:

1.  The author should reconfirm the formats of this manuscript. (1) The abstract should be a total of about 200 words maximum. The abstract should be a single paragraph and should follow the style of structured abstracts, but without headings(2) Abstract should not contain any references. (3) Results should contain a brief heading.

2.    The author should reconfirm the structures of this manuscript. Most of the text in the abstract should be in the section of the introduction, results, and discussion. Moreover, the author should cite an appropriate reference in the section of the introduction.

3.     The author should reconfirm the structures of a section in materials and methods (some description should be in the section of results). In materials and methods, the author should describe methods, protocolsreagents, etc. For example, the methods for SNP genotyping (real-time PCR?) and mediation analysis.

4.     I would suggest the author change some results of a table into a figure.

5.  The author should provide more convincing, novel, or differential results (especially in the microbiome) to show the clinical implications/critical findings in this study. The title includes the gut microbiome, which plays a critical role in metabolism and diabetes, especially for Akkermansia muciniphila. However, only a few results about the gut microbiota in this manuscript. Whole metagenome shotgun is a more powerful approach than 16S sequencing in the study of the gut microbiota; however, only 15 bacteria species were shown in this manuscript. I would suggest the author reconfirm the issue of the gut microbiota in this manuscript.

6.    The evidence in this manuscript is not enough to support the title of “mediate”. In discussion, the author should discuss (1) Lactate is strongly associated with diabetes and insulin resistance. (2) Gut microbiota (include Bacteroides massiliensis), lactate, and insulin sensitivity/resistance. (3) The clinical implications and causal relationship (SNP genotyping, ABO, metabolites, and the gut microbiota).

Reviewer 2 Report

Line 175 to 196:

The authors should include a complete description of the chromatographic methods used in this paper, including what columns were used (for both RP and HILIC methods), the mobile phases, modifiers, and gradient program. In addition, the authors should indicate the names of vendors of the LC and MS system used to acquire the data.

The list of internal standards (ISTD) that were used during LCMS analysis for both RP and HILIC analysis should be mentioned. What was the RSD% of the ISTDs throughout the LCMS analysis? Was any batch correction performed? How was the data normalized?

The authors should include the full list of the metabolites annotated by Metabolon (name, RT, and MS1 information).

The author should mention in the text that, in total, how many features were annotated by RP and HILIC MS?

Was method blank used during the LC-MS analysis? Was the blank subtraction performed after the data processing? If yes, briefly write about it in the materials and methods section.

What program was used to process the metabolomics data (both RP and HILIC)?

Include the PPM error of the 13 metabolites that were related to T2D.

Why didn't the authors use the whole metabolomics datasets (all the metabolites that were annotated) for their analysis? There might be some confounding factors that were not considered (for example, if smoke-related metabolites are detected among the participants of this study), or they might be some other biomarkers associated with T2D that were missing. Are the authors sure only these 13 plasma metabolites are related to T2D?

In line 48 for the following sentence (13 plasma metabolites that are related to T2D), include the references.

If you wanted to target 13 metabolites, why have you not performed targeted metabolomics instead of untargeted metabolomics to get the absolute quantity of the metabolites?

Table 2 for Beta estimates and 95% confidence intervals: (Beside reference box, also write: Beta (95% CI). The same for table 3

Reviewer 3 Report

Dear authors,

I revised the research article entitled ''The gut microbiome and plasma metabolites mediate differences in insulin homeostasis associated with ABO blood groups'', which I consider of interest to the scientific field. Still, I consider that some aspects must be improved in the current version of your manuscript, and you can see them in the following lines:

-abstract section: please remake the abstract section, into a maxim of 200 words. I suggest removing the Background details and briefly highlighting the aim, methods, and main results.

-line 79 (introduction):  please define shortly the ABO blood groups.

-introduction section: please add a few more examples of clinical studies involving the association of gut microbiota with ABO groups and T2D.

-line 139 (2.2. ABO haplotype and diplotype): How did you perform the genotyping? please mention the details.

-line 155 (2.3. Insulin homeostasis measurements): 75g of glucose represents the same quantity for both men and women? the same quantity for every particular body weight? Please clarify this.

Last but not least, there is no mentioned the ethical approval for the presented study, as it involved human subjects.